# Study on the Dynamic Mechanical Properties of Ultrahigh-Performance Concrete under Triaxial Constraints

**DOI:** 10.3390/ma16196591

**Published:** 2023-10-07

**Authors:** Wei Zhang, Jize Mao, Xiao Yu, Bukui Zhou, Limei Wang

**Affiliations:** 1College of Aerospace and Civil Engineering, Harbin Engineering University, Harbin 150001, China; 2Institute of Defense Engineering, Academy of Military Sciences, Beijing 100850, China

**Keywords:** ultrahigh-performance concrete (UHPC), dynamic impact, static triaxial constraint, strain rate effect, failure criterion

## Abstract

To confirm the effect of confining pressure on the dynamic mechanical behavior of ultrahigh-performance concrete (UHPC), this study used a true triaxial split Hopkinson pressure bar test system to perform dynamic compression tests on UHPC under triaxial constraints. The confining pressure range considered was 5~10 MPa, the strain rate range was 35~80 s^−1^, and the steel fiber contents were 0.5%, 1% and 2%. The three-dimensional dynamic engineering stress-strain relationship and equivalent stress-strain relationship of UHPC under different confining pressures and different strain rates were obtained and analyzed in detail. The results show that under the confinement condition, the dynamic peak axial stress–strain and dynamic peak lateral stress–strain of UHPC have strong sensitivity to the strain rate. In addition, the dynamic peak lateral stress–strain is more sensitive to the confining pressure than the dynamic axial stress. An empirical strength enhancement factor (DIFc) that considers the strain rate effect and confining pressure was derived, and the impact of the coupling between the enhancement caused by the confining pressure and the strain rate effect on the dynamic strength of the UHPC under triaxial confinement was discussed. A dynamic strength failure criterion for UHPC under triaxial constraint conditions was established.

## 1. Introduction

In current construction projects, concrete materials (normal reinforced concrete (NRC), mortar and ultrahigh-performance concrete (UHPC)) are still widely used. With the development of society and the increase in the global population, conflicts in some parts of the world are increasing. To protect the safety and property of the public, the dynamic behavior of concrete materials under impact loads caused by projectile impacts and explosions has attracted much attention. In such cases, the concrete material undergoes multiaxial compression with high confinement [1,2]. Therefore, the study of the mechanical response of concrete materials under triaxial compression not only is very important for the structural design of military and civil protection engineering projects, but also is conducive to the development and verification of constitutive models.

Researchers have performed many studies on the static response of concrete materials under triaxial compression conditions and have obtained many important conclusions. These studies are helpful for understanding the effect of constraint conditions on concrete strength, deformation capacity and failure mechanisms. Wang et al. [3] conducted a series of triaxial compression tests on a 100 mm cube test specimen with an unconfined compressive strength of approximately 10 MPa and obtained an approximately linear relationship between the normalized triaxial compressive strength and the constraint ratio. Noori et al. [4] prepared two kinds of steel fiber-reinforced mortar and ordinary high-performance concrete cylindrical specimens with 1% and 2% steel fiber volume fractions and conducted triaxial compression tests. The results show that the increase in confining pressure increases the peak value of axial stress-strain, making the test specimen exhibit stronger ductility. The addition of steel fibers is beneficial to improve the energy absorption capacity of these materials. Ren et al. [5] and Lu et al. [6] obtained a triaxial stress–strain relationship through triaxial compression tests on UHPC, and discussed the failure criterion and toughness of UHPC under triaxial compression. Chi et al. [7] developed a failure envelope based on the five-parameter failure criterion through a true triaxial compression test of hybrid steel–polypropylene fiber-reinforced concrete (HFRC). The proposed envelope curve can provide accurate approximations of the ultimate strength for plain concrete and fiber-reinforced concrete under static loading. Sirijaroonchai et al. [8] studied various fiber types (high-strength hooked steel fibers and ultrahigh-molecular-weight polyethylene fibers) and fiber volume fractions (1~2%) through passive triaxial testing of high-performance fiber-reinforced cement-based composites under different confining pressures (41 MPa and 52 MPa). The results showed that for a higher confinement ratio, the confinement effect introduced by various types and volume fractions of fibers decreases. Jiang et al. [9] conducted triaxial compression tests under passive confinement on cube specimens of normal reinforced concrete (NRC) by means of a newly developed triaxial test system. Compared to the traditional triaxial compression calibrated model, the peak strength prediction error was within ±10%. These results provided new ideas for studying the failure mechanism of concrete under passive confinement.

In addition, Imran et al. [10], Vu et al. [11] and Piotrowska et al. [12] systematically studied the influence of water content, fiber, aggregate and cement paste on the triaxial compressive properties of concrete.

However, to date, the dynamic response of concrete under triaxial confinement has seldom been discussed [13], mainly because it is still challenging to perform tests by coupling confining pressure and dynamic loading. At present, based on traditional split Hopkinson pressure bar (SHPB) test technology, there are two methods to achieve triaxial compression: namely, the active confining pressure method and the passive confining pressure method. The active confining pressure loading system applies radial confining pressure to the specimen before dynamic loading. Gran et al. [14] studied the dynamic behavior of concrete when considering confinement using an active confining pressure device. The results showed that, compared with the quasistatic condition, the strain rate effect was not observed when the strain rate was 0.5 s^−1^. However, when the strain rate reached 1.3~5 s^−1^, the shear failure envelope established by the triaxial compression data was 30~40% higher than that under static loading. Zeng et al. [15] used an MTS device to obtain a series of complete stress-strain curves for specimens subjected to different strain rates and confining pressure combinations. The results confirmed that there was a significant coupling effect between the strength enhancements controlled by the strain rate and confining pressure. Based on the 1D SHPB test system, Xue and Hu [16] precharged the cement mortar test specimen with a constant liquid confining pressure through hydraulic cylinders and found that the strain rate effect was significant for cement mortar under confining pressure. Similarly, Marvern et al. [17] and Gary et al. [18] applied confining pressures of 3~10 MPa to concrete specimens. The experiments found that the concrete specimen strength had strain rate sensitivity under this range of confining pressures. Under a confining pressure of 5 MPa, the strength of concrete increased by 30% when the strain rate increased from 250 s^−1^ to 600 s^−1^. Yan et al. [2] and Fujikake et al. [19] implemented a series of triaxial dynamic compression tests of concrete under low-strain-rate loading conditions (<2 s^−1^) and found that with an increasing confining pressure, the strain rate effect was weakened. When the confining pressure exceeded the uniaxial compressive strength of concrete, the strain rate sensitivity of the concrete basically disappeared. Fu et al. [20] studied the dynamic compression characteristics of hybrid fiber-reinforced concrete under confining pressure and established an empirical dynamic strength criterion by using the active confining pressure applied via the SHPB test. The passive confining pressure method is a relatively new technology that has been widely used in recent years. In this method, a lateral ring is added around the sample to limit its expansion, thus generating a constraining force, which is the lateral confining pressure [11]. Liu et al. [21] used an SHPB dynamic loading device to investigate the dynamic mechanical properties of NRC under constraint conditions. The confining pressure was obtained by imposing a transverse metal sleeve on the axially loaded specimen. The test results show that the dynamic peak axial stress, dynamic peak lateral stress and peak axial strain of concrete are very sensitive to the strain rate under constraint conditions. Li et al. [22] simulated the SHPB test of concrete materials using the finite element method and the Drucker–Prager (DP) model. They found that due to the limitation of the lateral inertia effect, the apparent strength increase in the concrete material was due to the influence of the hydrostatic pressure rather than the strain rate sensitivity of the material.

Notably, in the active confining pressure method, the pressure (σ_2_ = σ_3_) is considered to be a constant value, but it is difficult to simultaneously measure the changes during the impact loading process, and the hydraulic oil is prone to leak under a high confining pressure [15,18]. Under passive constraints, the confining pressure depends on the gap between the specimen and the sleeve and the material properties and thickness of the sleeve. It is relatively difficult to precisely control the gap between the specimen and the sleeve, and the lateral stress varies during the dynamic loading process [23].

Researchers have made enormous efforts to obtain the true triaxial stress state of the test specimen and the stress response state in different directions during the dynamic impact process. Cui et al. [24] used a newly developed three-dimensional SHPB (3D-SHPB) device to simultaneously compress a test specimen with the same amplitude in three mutually perpendicular directions to obtain the volume characteristics of concrete under dynamic hydrostatic pressure. Liu et al. [25] conducted a series of experiments on cement and concrete specimens using the 3D-SHPB loading system to investigate the mechanical properties and fracture behavior of concrete-based materials under coupled static-dynamic loading conditions. The test results showed that the value of the dynamic uniaxial compressive strength of concrete was much higher than the static uniaxial compressive strength, but that it decreased with axial prestress. The dynamic compressive strength under a triaxial constraint is approximately 3 times higher than the quasistatic uniaxial strength. Xu et al. [26] used their own developed dynamic test system under static triaxial constraints to investigate the dynamic compressive performance of NRC specimens under different static triaxial constraint conditions. The three-dimensional dynamic engineering stress–engineering strain relationship, dynamic volumetric strain–hydrostatic pressure relationship and equivalent stress–equivalent strain relationship were obtained and analyzed in detail. The findings indicated that the load path dependence and the strain rate dependence were significant. Liu et al. [27] carried out a series of dynamic and static compression tests on NRC with the 3D-SHPB system. It was concluded that there was an obvious strain rate effect on NRC when it was under triaxial confining pressure. The confining pressure has an effect on the dynamic stress of concrete, but with the increase in strain rate, the effect gradually decreases.

In summary, the mechanical response of concrete materials under triaxial confining pressure that has been mostly focused on is the quasistatic mechanical response. Due to the limitations of the test conditions, only a few dynamic tests under triaxial confining pressure have been carried out at present. The research target has mostly been NRC. However, the application of UHPC in the field of modern protective engineering is increasing. Research on the dynamic mechanical response of UHPC under triaxial compression is still lacking. In this work, dynamic mechanical tests of UHPC under triaxial confining pressure were carried out by using a true triaxial split Hopkinson pressure bar system. Considering the influence of the confining pressure stress and strain rate, the engineering stress-strain and equivalent stress-strain relationships of UHPC were analyzed. The dynamic strengthening mechanism of UHPC under triaxial constraints was discussed, and the dynamic failure criterion was fitted based on the test results.

## 2. True Triaxial Test

### 2.1. Sample Materials

The UHPC used in this paper was composed of Chinese standard grade 52.5 P.II Portland cement, silica fume, fine aggregate and a water reducer, and the mix ratios are shown in Table 1. The steel fibers were copper-plated straight steel fibers, and the volume additions were 0.5%, 1% and 2%. The fiber length, diameter and tensile strength were 12 mm, 0.2 mm and 2850 MPa, respectively.

The size of the SHPB test specimen used in this experiment was 50 mm × 50 mm × 50 mm, as shown in Figure 1. To ensure the smoothness of the test specimen and reduce the stress concentration during the experiment, the two end faces of the test specimen were polished after curing so that the parallelism deviation of the three opposite faces was less than 0.25 mm, and the dimensional accuracy of the test specimen was kept within ±0.5%. The end-face flatness deviation was less than 0.05 mm, the adjacent surfaces of the cubic test specimen had good verticality and the maximum deviation was less than 0.3°. To reduce the discreteness of the test results, 2 specimens were tested for each confining pressure and each strain rate, for a total of 54 specimens.

In addition, a total of nine 100 mm × 100 mm × 100 mm cube test specimens were prepared. The average strength of three specimens of each type was taken as the quasistatic compressive strength of each type of UHPC material. The measured strength and average strength are shown in Table 2.

### 2.2. Triaxial Test

#### 2.2.1. Test Equipment

This work adopted a true triaxial split Hopkinson pressure bar test system, and a diagram of the test setup is shown in Figure 2.

The experimental system included two parts: ① A true triaxial static load application system, which was composed of hydraulic cylinders ((4), (8) and (11) in Figure 2) that were oriented in three orthogonal directions, and their corresponding reaction supports, which could test the cube compressive stress in three directions. ② A bullet launching and signal test system, which was mainly composed of a high-pressure gas cannon (1) in the impact direction (x-direction); an incident square rod (2) and a supporting square rod (3) horizontal to the impact direction; a supporting square bar ((6) and (7)) in the y-direction; and a supporting square bar ((9) and (10)) in the z-direction, where the y- and z-directions are perpendicular to the impact direction. The experiment console (12) controls the hydraulic system and the launch system. The hydraulic station (13) provides hydraulic oil for the three hydraulic cylinders in the servo control process.

As shown in Figure 3, to reduce the loss of equipment, a gasket made of the same material as the steel rod was added to the surface of the specimen, and its surface was evenly coated with Vaseline to reduce the interface friction [28]. During the test, the specimen is placed at the intersection of the six bar axes. The incident rod, specimen, transmission rod and servo-controlled hydraulic cylinder are constrained by the x-axis steel frame, and the static constraint in the x-axis direction (σ_x-static_) can be realized by pumping the hydraulic cylinder. A cylindrical impactor impacts the incident square rod through the predesigned hole in the steel frame. The cylindrical absorbing rod passes through the hole in the hollow servo-controlled hydraulic cylinder, where it makes contact with the transmission rod and transfers a part of the transmitted waves to the momentum trap. The static constraints (σ_y-static_ and σ_z-static_) in the y-axis and z-axis directions are loaded in the same way as that in the x-axis direction, both by pumping hydraulic cylinders. The diameter of the cylindrical impacting rod was 42 mm, and the length was 500 mm. The lengths of the incident rod and the transmission rod were 2500 mm and 2000 mm, respectively. Their cross-sections were squares of 50 mm × 50 mm. The diameter of the cylindrical absorption rod was 42 mm, and the length was 800 mm. In this study, the lengths of the left and right rods along the y-axis were both 2000 mm, and their cross-sections were squares of 50 mm × 50 mm; the lengths of the upper rod and the lower rods along the z-axis are both 1600 mm, and their cross-sections were also squares of 50 mm × 50 mm [26].

#### 2.2.2. Testing Techniques

In this work, the difference between the traditional SHPB test and 3D-SHPB test was that the 3D-SHPB test was mainly divided into two stages. In the first stage, the static load was preloaded, the six surfaces of the cube were in contact with six square steel rods, and then the load was applied to the test specimen through the hydraulic device and six steel rods. The static loads were σ_x-static_, σ_y-static_ and σ_z-static_. In the second stage, a dynamic load was applied, and the impacting rod was launched in the x-direction. The impact of the impacting rod on the incident rod generated elastic compression waves, named incident waves (*ε_i_*), which propagate toward the specimen. When the incident wave reached the incident rod/specimen interface, due to the impedance mismatch between the incident rod/transmission rod and the test specimen, the incident wave was divided into the reflected wave *ε_r_* and the transmitted wave *ε_t_*, and the reflected wave was propagated in the opposite direction of the incident rod. The transmitted waves passed through the specimen and entered the transmission rod, which was also the cause of the plastic deformation of the test specimen. At the same time, due to the dynamic load applied to the x-axis, the test specimen experienced lateral expansion (i.e., the Poisson effect). Therefore, a wave was measured with the four square bars in the y-axis and z-axis directions: *ε_y-left_, ε_y-right_, ε_z-up_* and *ε_z-down_*.

At the same time, the basic principle of the true triaxial split Hopkinson pressure bar system is the same as that of the traditional SHPB test technique, and the theory of one-dimensional wave propagation in an elastic bar also needs to be followed. Therefore, the stress along the x-axis $\sigma_{x-dyn}$, the corresponding strain $\varepsilon_{x-dyn}$ and the corresponding strain rate ${\overset{\cdot }{\varepsilon }}_{{}_{x-dyn}}$ are still calculated using the three-wave method [24]:(1)$$\sigma_{x-dyn}(t)=\frac{P_{s-x}}{A_{s-x}}=\frac{EA_{0}}{2A_{s-x}}\left[\varepsilon_{i}(t)+\varepsilon_{r}(t)+\varepsilon_{t}(t)\right]$$
(2)$$\varepsilon_{x-dyn}(t)=\frac{C_{0}}{l_{s-x}}\int_{0}^{t}\left[\varepsilon_{i}(t)-\varepsilon_{r}(t)-\varepsilon_{t}(t)\right]dt$$
(3)$${\dot{\varepsilon }}_{x-dyn}(t)=\frac{d\varepsilon_{x-dyn}(t)}{dt}=\frac{C_{0}}{l_{s-x}}\left[\varepsilon_{i}(t)-\varepsilon_{r}(t)-\varepsilon_{t}(t)\right]$$
where *C*_0_, *E* and *A*_0_ are the wave velocity, elastic modulus and cross-sectional area of the pressure bar, respectively. *P_s−x_, l_s−x_* and *A_s−x_* are the pressure in the x-axis direction and the length and area of the sample along the x-axis, respectively. *ε_i_*, *ε_r_* and *ε_t_* are the incident wave, reflected wave and transmitted wave, respectively.

For the y-axis and z-axis directions, the particle velocity and stress state at both ends of the test specimen can be directly obtained from the strain gauge signals on the square bar. Therefore, the stress, strain and strain rate along the y-axis and z-axis are calculated as follows [29]:(4)$$\sigma_{k}(t)=\frac{A_{0}}{2A_{k}}E_{0}\left|\varepsilon_{k,1}(t)+\varepsilon_{k,2}(t)\right|$$
(5)$$\varepsilon_{k}(t)=-\frac{C_{0}}{L_{k}}\int_{0}^{t}\left|\varepsilon_{k,1}(t)+\varepsilon_{k,2}(t)\right|dt$$
(6)$${\dot{\varepsilon }}_{k}(t)=\frac{\left|\Delta V\right|}{L_{k}}=\frac{C_{0}\left|\varepsilon_{k,1}(t)+\varepsilon_{k,2}(t)\right|}{L_{k}}$$
where k represents the y-axis or z-axis and L_k_ and A_k_ are the length and area of the sample in the k-direction (y-axis or z-axis), respectively. Δv is the velocity difference between opposing surfaces in the k-direction. When k represents the y-axis, $\varepsilon_{k,1}$ and $\varepsilon_{k,2}$ are $\varepsilon_{y,left}$ and $\varepsilon_{y,right}$, and when k represents the z-axis, they are $\varepsilon_{z,up}$ and $\varepsilon_{z,down}$. The other parameters are the same as in Equations (1)~(3).

## 3. Test Results and Analysis

Notably, the application condition of the above equations is that the stress in the x-direction in the specimen should reach the equilibrium state, and the two waves in the y-direction and the two waves in the z-direction both have good consistency, which will be discussed in Section 3.1.

### 3.1. Test Waveform Analysis

Before the dynamic test, a stress equilibrium check must be performed [30]. Formulas (7)~(9) can be used to check the stress equilibrium state of the test specimen.
(7)$$\sigma_{x-t}=\sigma_{x-i}+\sigma_{x-r}$$
(8)$$\sigma_{y-left}=\sigma_{y-right}$$
(9)$$\sigma_{z-up}=\sigma_{z-down}$$
where $\sigma_{x-t}$, $\sigma_{x-i}$ and $\sigma_{x-r}$ represent the transmitted stress, incident stress and reflected stress of the test specimen in the x-axis direction, respectively; $\sigma_{y-left}$ and $\sigma_{y-right}$ represent the dynamic stress on the left and right sides of the test specimen in the y-axis direction, respectively; and $\sigma_{z-up}$ and $\sigma_{z-down}$ represent the dynamic stress on the upper and lower sides of the test specimen in the z-axis direction, respectively.

Figure 4 shows a typical stress equilibrium examination performed by U0.5 when the triaxial static pressure is 5 MPa and impact velocity V = 13 m/s. The test specimen reached the stress equilibrium, which confirmed the validity of the dynamic mechanical property test results under true triaxial static loading in the present study.

### 3.2. Dynamic Stress-Strain Relationship

Table 3 lists the dynamic mechanical properties of UHPC under triaxial compression. The strain rates in the table are the strain rates in the x-axis direction. It can be seen from the table that the dynamic compressive behavior of UHPC with three fiber dosages was very sensitive to the strain rate. Under the same fiber dosage, the peak stress of UHPC in the x-axis direction increased significantly with an increasing strain rate. The increase in confining pressure has less of an effect on the strength in the x-axis direction but has a greater impact on the strength in the y-axis and z-axis directions. The dynamic response of the U0.5 test specimen under triaxial confining pressure was taken as an example for detailed analysis.

Figure 5a shows the typical experimental results of the specimen under triaxial confining pressure. It can be seen from the figure that the dynamic response of the specimen under confining pressure goes through five stages. In the first stage, the stress–strain curve is slightly concave, indicating that the internal voids of the concrete are compacted under loading. However, it should be noted that the dynamic response of this stage will be different at high strain rates, as there is no obvious compression stage at high strain rates [31]. In the second stage, the stress increases linearly with strain. In the third stage, the growth trend of stress slows with increasing strain, probably because the existence of the confining pressure limits the free development of internal cracks in the specimen and increases the energy dissipation rate of concrete. In the fourth stage, the stress reaches the peak value, and with the increase in strain, the stress does not increase, showing an obvious plastic stage. In the fifth stage, with the further increase in strain, the stress decreases suddenly. Compared with the traditional 1D SHPB test, the strain softening behavior in this study was not obvious. Figure 5b shows the corresponding typical strain time-history curve. It can be seen that the strain increased rapidly with the loading time until the peak strain was reached. With continuous loading, the peak strain increased slowly until the specimen failed.

Figure 6 shows the dynamic stress-strain relationship along the x-axis direction under the action of triaxial confining pressure. During the test, the strain rate range was 35~80 s^−1^, and the static load was in the range of 5~10 MPa. The test results show that under the same static load, the dynamic stress and dynamic strain of the test specimen in the x-axis direction increased with an increasing strain rate; for example, when [σ _x-static_, σ _y-static_, σ _z-static_] = [5, 5, 5], as the strain rate increased from 35 s^−1^ to 80 s^−1^, the dynamic stress increased from 138 MPa to 264.4 MPa, and the dynamic strain increases from 0.005 to 0.01, showing a significant strain rate effect, which was also consistent with the results of the traditional 1D SHPB test. Under the same strain rate, as the lateral confining pressure increased, the dynamic stress-strain in the x-axis direction increased slightly, but the effect was insignificant.

Figure 6a shows that under the same confining pressure, with the increase in the strain rate the elastic modulus increased slightly, which was consistent with the conclusion of the 1D SHPB test. At the same strain rate, the increase in confining pressure had a small effect on the elastic modulus, which may be due to the limitations of the experimental conditions. The strain rate range and the preset confining pressure range in the study were narrow, resulting in similar test results. Reference [21] suggested that the confining pressure limits the deformation of concrete to improve the energy dissipation rate of concrete under compression, improve the deformation ability of concrete and reduce the deformation modulus. A follow-up study will further analyze and discuss the relationship between the elastic modulus and the size of the preloaded pressure through simulation research.

The impact load was applied to the test specimen along the x-axis. Due to the dynamic Poisson effect, deformation of the test specimen along the y-axis and z-axis was inevitable. Figure 6b,c show the dynamic response of the y-axis and z-axis, respectively, when the impact velocity was 30 m/s; that is, the strain rate was about 80 s^−1^. With the increase in the static load and the confining pressure, the lateral stress-strain of the test specimen increased. Compared to the dynamic response on the x-axis, the dynamic stress and strain of the y-axis and z-axis are more sensitive to the static load of the confining pressure. For example, when the static load of the confining pressure [σ _x-static_, σ _y-static_, σ _z-static_] = [5, 5, 5] increased to [σ _x-static_, σ _y-static_, σ _z-static_] = [10, 10, 10], the peak stress along the y-axis increased from 15 MPa to 22.5 MPa, an increase of approximately 50%, and the peak strain increased from 0.0004 to 0.0008, an increase of approximately 100%. Since the static load was the same σ _y-static_ = σ _z-static_, the stress-strain relationship of the z-axis was similar to that of the y-axis.

### 3.3. Relationship between Dynamic Equivalent Stress and Equivalent Strain

The equivalent stress and equivalent strain are [32]
(10)$$\bar{\sigma }=\sqrt{\frac{1}{2}\left[(\sigma_{x}-\sigma_{y}{)}^{2}+{\left(\sigma_{y}-\sigma_{z}\right)}^{2}+{\left(\sigma_{z}-\sigma_{x}\right)}^{2}\right]}$$
where $\sigma_{x}=\sigma_{x-dyn}+\sigma_{x-static}$, $\sigma_{y}=\sigma_{y-dyn}+\sigma_{y-static}$ and $\sigma_{z}=\sigma_{z-dyn}+\sigma_{z-static}$, and
(11)$$\bar{\varepsilon }=\sqrt{\frac{2}{9}\left[(\varepsilon_{x}-\varepsilon_{y}{)}^{2}+{\left(\varepsilon_{y}-\varepsilon_{z}\right)}^{2}+{\left(\varepsilon_{z}-\varepsilon_{x}\right)}^{2}\right]}$$
where $\varepsilon_{x}=\varepsilon_{x-dyn}+\varepsilon_{x-static}$, $\varepsilon_{y}=\varepsilon_{y-dyn}+\varepsilon_{y-static}$ and $\varepsilon_{z}=\varepsilon_{z-dyn}+\varepsilon_{z-static}$.

Figure 7 shows the relationship curve of the equivalent stress and effective strain of the U1 test specimen under different strain rates when the hydrostatic pressure was [10, 10, 10]. Similar to the engineering stress-strain relationship, the equivalent stress-strain was roughly divided into five stages, namely, the compaction stage, linear increase stage, slow increase stage, plastic stage and stress unloading stage. As the impact velocity increased from 13 m/s to 30 m/s and the strain rate increased from 30 s^−1^ to 80 s^−1^, the slope before the peak was almost unchanged, while the equivalent peak strength increased from 134.7 MPa to 266.95 MPa, an increase of 98.2%; additionally, the equivalent peak strain increased from 0.0031 to 0.0085, an increase of 174.2%. This exhibited a clear strain rate effect.

Figure 8 shows the relationship between the equivalent stress and the effective strain of the U1 specimen when the strain rate was 60 s^−1^ and the confining pressure increased from 5 MPa to 10 MPa. The equivalent peak strength increased from 177.5 MPa to 178.2 MPa, an increase of 0.3%; the equivalent peak strain increased from 0.0046 to 0.0048, an increase of 4.3%. This shows that when the strain rate was 30~100 s^−1^, the effect of the confining pressure on the equivalent stress-strain was insignificant. This was consistent with the study by Xu et al. [26] on the dynamic strength response of normal strength concrete under triaxial confining pressure.

### 3.4. Dynamic Strength Enhancement Factor

Based on the traditional 1D SHPB test, the dynamic increase factor (DIF) of strength is defined as the ratio of the dynamic strength to the quasistatic strength of the test specimen under uniaxial compression, and it is widely used to analyze the mechanical response of material under dynamic impact loading [22,33]. Notably, the dynamic mechanical properties of traditional concrete only use the dynamic strength in the impact direction, and it is difficult to reflect the influence of static loading. Therefore, the test data in this paper were analyzed by using the equivalent strength enhancement factor (DIFc), which can be expressed as:(12)$$DIF_{C}=\bar{\sigma }/\sigma_{c}$$
where $\bar{\sigma }$ is the dynamic equivalent strength and $\sigma_{c}$ is the unconfined quasistatic compressive strength of concrete. The dynamic equivalent strength factor DIFc of UHPC measured in this experiment under the action of true triaxial confining pressures of 5~10 MPa is plotted in Figure 9. As shown in Figure 9, as the strain rate increased, the DIFc increased rapidly. At the same rate, the DIFc was significantly higher than the DIF value under conventional SHPB compression.

This indicates that the enhancement of concrete strength under the coupling action of confining pressure and strain rate is greater than the enhancement effect of a single strain rate effect.

For concrete-like materials, several empirical formulas have been proposed based on conventional SHPB tests to estimate the influence of the strain rate effect on compressive strength, and the results are summarized in Table 4. Based on the data studied in this paper, we fitted the DIFc. Due to the lack of test data for UHPC under dynamic loading and confining pressure, it is difficult to accurately determine the transition strain rate in this study, so we refitted the DIFc based on the CEB model. The empirical formula of the DIFc is shown in Table 4, and the CEB model specification is used for the strain rates lower than the transition strain rate.

## 4. Discussion

### 4.1. Analysis of the Dynamic Enhancement Mechanism under Confining Pressure

Many experimental and theoretical studies have been conducted on the sensitivity of concrete materials to the loading rate [36,37]. The physical mechanisms of the strain rate effect can be classified into three aspects [21]: the thermal activation mechanism, viscosity mechanism (Stefan effect) and inertia mechanism. Additionally, studies have shown that there is a coupling effect between the enhancement caused by the confining pressure and the strain rate effect. However, there is still no consensus on the mechanism of the concrete strain rate effect when considering the confining pressure.

The thermal activation mechanism causes a thermal vibration effect on an atomic scale. Thermal vibration will break the atomic bonds, resulting in microcracks. The stronger the dynamic load is, the more microcracks that form. When a test specimen is in the low strain rate range, the failure of the concrete is mainly caused by the development of a small number of cracks, and the cracks will develop along the weakest path. With an increasing strain rate, the stress wave cannot be transmitted out of the specimen in a short time, and many microcracks will be generated in the test specimen to dissipate this energy. Therefore, macroscopically, the increase in dynamic strength is small under a low strain rate, and the increase in dynamic strength is large at a high strain rate. When confining pressure exists in the test specimen, the magnitude of the strength increase caused by the strain rate effect is reduced because the confining pressure limits the development of microcracks, which is similar to the conclusion of Liu et al. [21]; when the strain rate increases, the magnitude of the strength increase caused by the strain rate effect is reduced. Due to the change in the energy consumption mode, this weakening effect gradually decreases.

The viscosity mechanism is related to the water content of concrete and can be simply summarized as follows: there is a thin layer of viscous film (free water) between the concrete matrix units on both sides of the microcrack, with a certain distance h between them. When two matrix units are separated by velocity v, an opposing force is generated [38]. The greater the speed, the greater the opposing force. In the compression test performed in this work, the separation and slip of the surface parallel to the inclined crack were considered, as shown in Figure 10. It can be seen from the figure that in the test specimen with added confining pressure, the existence of the confining pressure reduced the slip rate of the concrete matrix elements on both sides of the microcrack, thus weakening the dynamic enhancement of compressive strength.

Researchers generally believe that the inertia effect mainly controls the dynamic strength of concrete under a high strain rate. When concrete is in the high strain rate range, the macroscopic bearing capacity of a concrete material increases with the increase in strain rate, while the true dynamic strength has a limit value at a very high strain rate. In the high strain rate range, the concrete can still bear a load, but ultimately the concrete will fail after unloading. Therefore, the inertia effect was not considered within the scope of the present study.

In summary, when the strain rate is low, the strength increase caused by the strain rate effect is small, while the strength increase caused by the confining pressure is the main reason for the improvement in concrete strength. With an increasing strain rate, the increase in strength caused by the strain rate effect becomes the main reason for the improvement in concrete strength, as shown in Figure 11. However, the contribution of the confining pressure and strain rate effects to the increase in strength at different stages needs to be further study. In addition, in the dynamic impact process, the confining pressure will have a negative effect on the strength increase caused by the strain rate effect, but with the increase in strain rate, this negative effect will gradually weaken.

### 4.2. Failure Criterion

According to the three-parameter failure criterion proposed by Bresler–Pister in an octahedral stress space [39], the failure envelope surface of this failure criterion tends to be a quadratic surface, which is smooth and convex in shape. The model expression is as follows:(13)$$\tau_{0}=a-b\sigma_{0}+c\sigma_{0}^{2}$$
where a, b and c are material parameters, which together determine the shape and size of the failure envelope. To be suitable for general situations, normalization is performed by the corresponding uniaxial compressive strength fc to determine the failure surface parameter. By fitting the normalized compression test data, the UHPC failure surface parameters related to the uniaxial compressive strength are obtained. And, $\tau_{0}$ and $\sigma_{0}$ are calculated as follows:(14)$$\tau_{0}=\frac{\tau_{oct}}{f_{c}};\;\sigma_{0}=\frac{\sigma_{oct}}{f_{c}}$$
where
(15)$$\tau_{oct}=\frac{\sqrt{(\sigma_{x}-\sigma_{y}{)}^{2}+{\left(\sigma_{y}-\sigma_{z}\right)}^{2}+{\left(\sigma_{z}-\sigma_{x}\right)}^{2}}}{3};\;\sigma_{oct}=\frac{\sigma_{x}+\sigma_{y}+\sigma_{z}}{3}$$

To consider the effect of the strain rate on the failure criterion of concrete under true triaxial dynamic loading, Equation (13) is rewritten as:(16)$$\tau_{0}=a_{{}_{\overset{•}{\varepsilon }}}-b_{{}_{\overset{•}{\varepsilon }}}\sigma_{0}+c_{{}_{\overset{•}{\varepsilon }}}\sigma_{0}^{2}$$
where $a_{\overset{•}{\varepsilon }}$, $b_{\overset{•}{\varepsilon }}$ and $c_{\overset{•}{\varepsilon }}$ are the rate effect parameters and $\tau_{0}$ and $\sigma_{0}$ are calculated according to Equations (14) and (15). Equation (16) was used to perform regression analysis on the test data, and the obtained rate effect parameter values are shown in Table 5. The relationship of the corresponding values $\tau_{0}$~$\sigma_{0}$ is shown in Figure 12.

## 5. Conclusions

The dynamic compression test of UHPC was conducted using a true triaxial SHPB test system setup. The strain rates were in the range of 35~80 s^−1^, and the confining pressure was 5~10 MPa. Based on the results of the dynamic compression test, the dynamic strength, equivalent stress-strain, DIFc, enhancement mechanism and failure criterion of UHPC were investigated. The main conclusions are as follows:The peak stress, peak strain, equivalent peak stress and equivalent peak strain of UHPC increase obviously with an increasing strain rate in the x-axis loading direction. The confining pressure has little influence on the dynamic response in the x-axis direction, but has a greater influence on the stress and strain in the y-axis and z-axis directions.The equivalent strength enhancement factor DIF_c_ of UHPC under confining pressure is established and fitted. Under the same strain rate, the equivalent strength, DIF_c_, is larger than the DIF obtained from the 1D SHPB test. Based on this, the empirical formula of the DIF_c_ of UHPC under confining pressure is fitted.There is a coupling effect between the enhancement caused by the confining pressure and the strain rate effect. When the strain rate is low, the extent of the dynamic strength increase caused by the strain rate effect is small, and the strength increase caused by the confining pressure is the main reason for the increase in concrete strength. As the strain rate increases, the weakening effect of the confining pressure gradually weakens, and the strength increase caused by the strain rate effect becomes the main reason for the increase in concrete strength.An improved three-parameter dynamic failure criterion is established and calibrated for this failure criterion.

## Figures and Tables

**Figure 1 materials-16-06591-f001:**
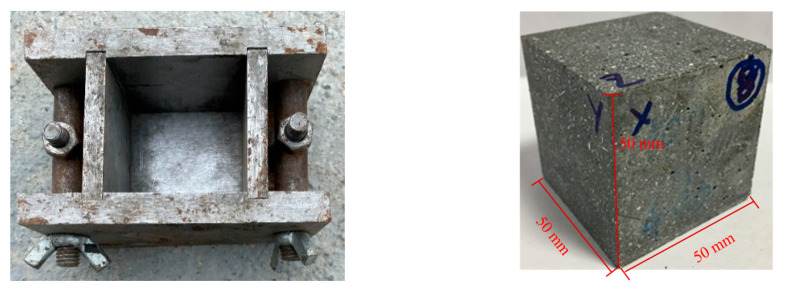
Trial mold and test specimen.

**Figure 2 materials-16-06591-f002:**
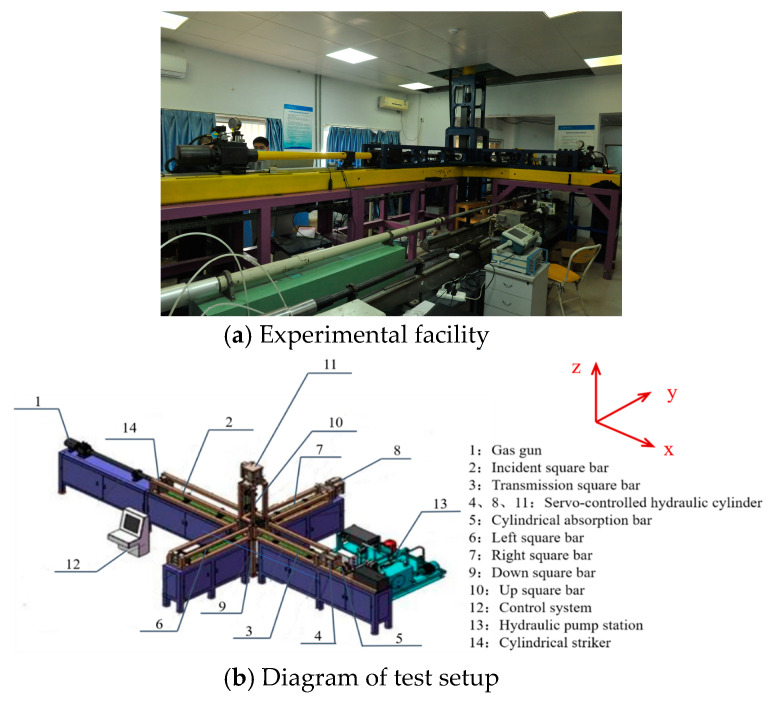
The true triaxial split Hopkinson pressure bar loading system.

**Figure 3 materials-16-06591-f003:**
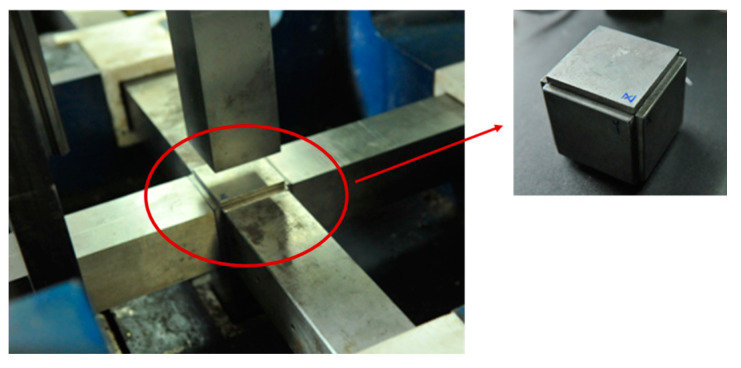
Standard cubic specimen placed between the bars of the loading element.

**Figure 4 materials-16-06591-f004:**
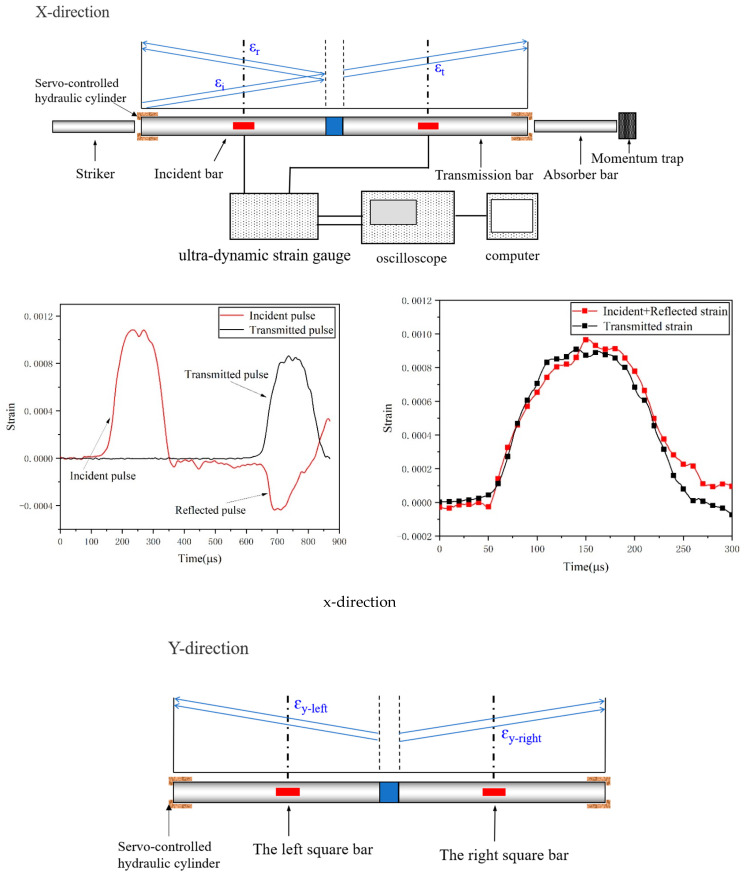
Stress equilibrium.

**Figure 5 materials-16-06591-f005:**
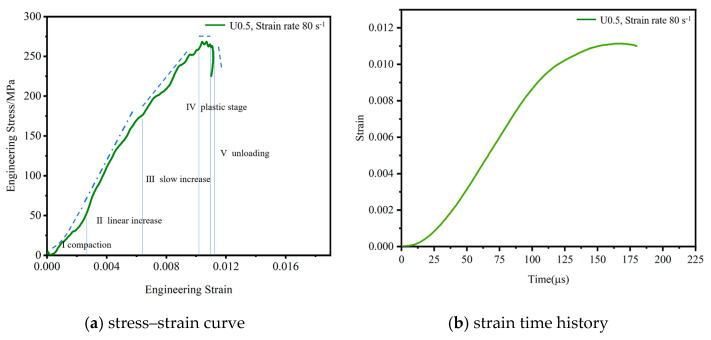
Typical curve in x-direction.

**Figure 6 materials-16-06591-f006:**
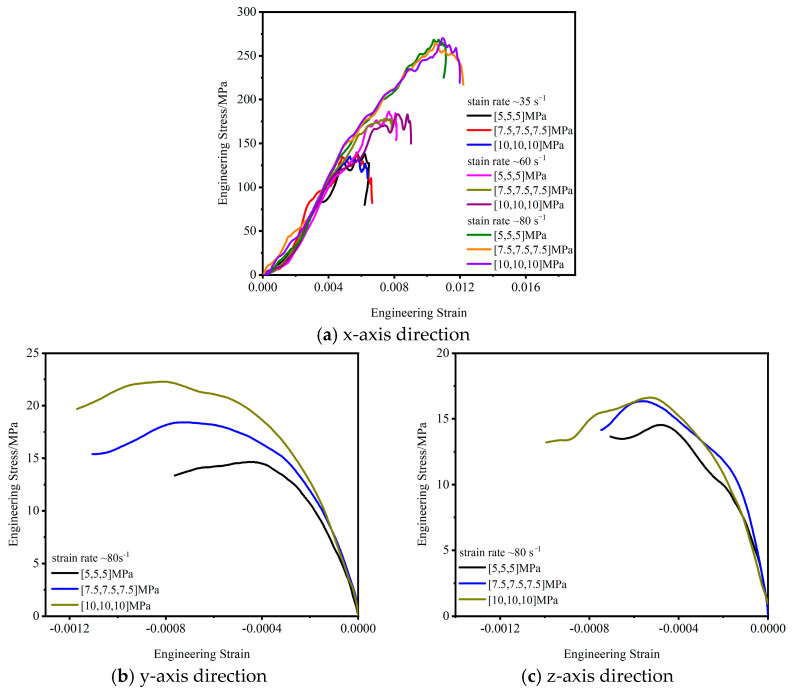
Stress-strain curve.

**Figure 7 materials-16-06591-f007:**
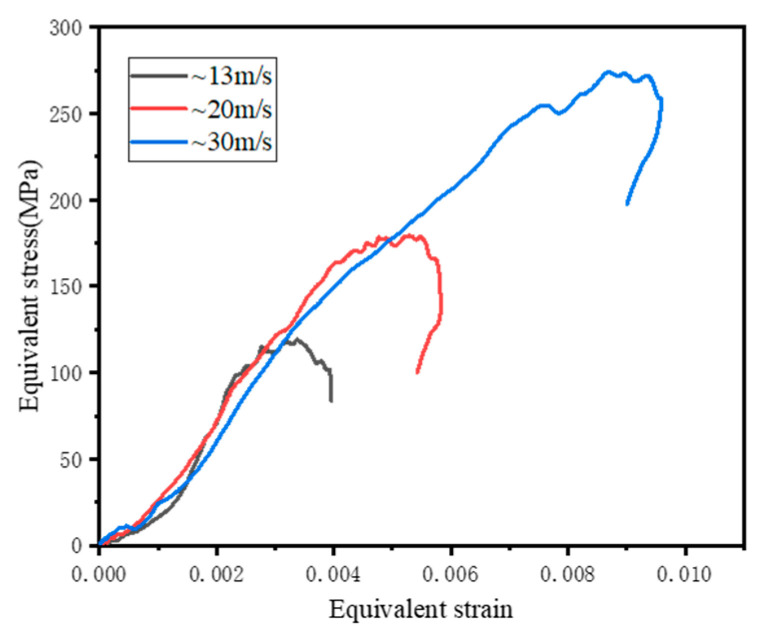
Equivalent stress-strain curves of U1 under different strain rates.

**Figure 8 materials-16-06591-f008:**
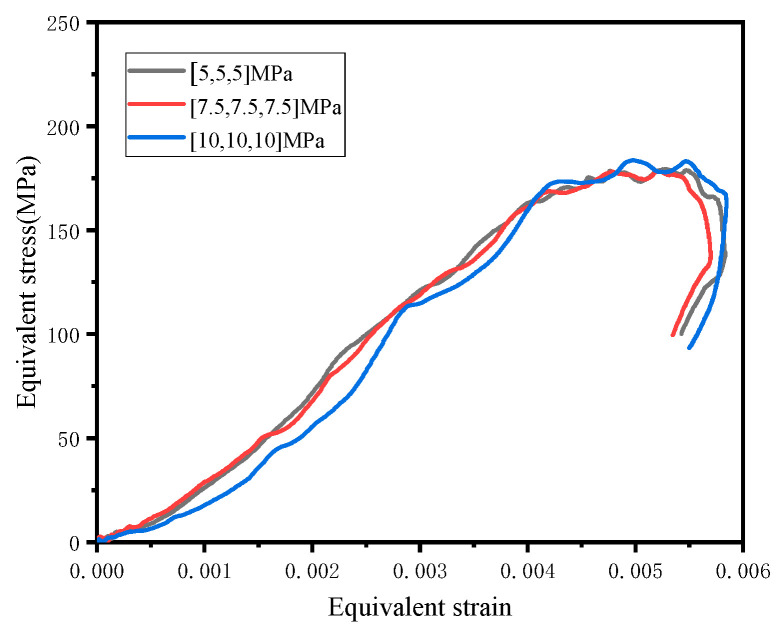
Equivalent stress-strain curve of U1 when the strain rate was 60 s^−1^.

**Figure 9 materials-16-06591-f009:**
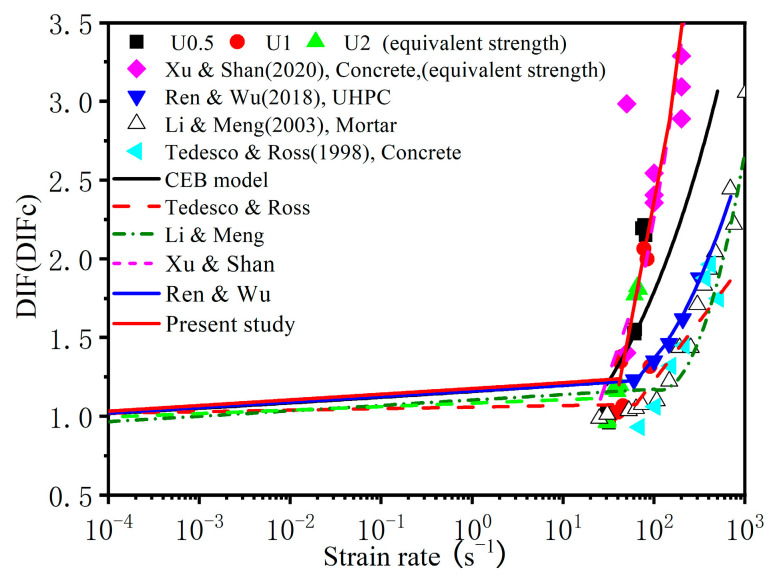
DIF (DIF_c_) for the compressive strength of UHPC [22,26,33,34,35].

**Figure 10 materials-16-06591-f010:**
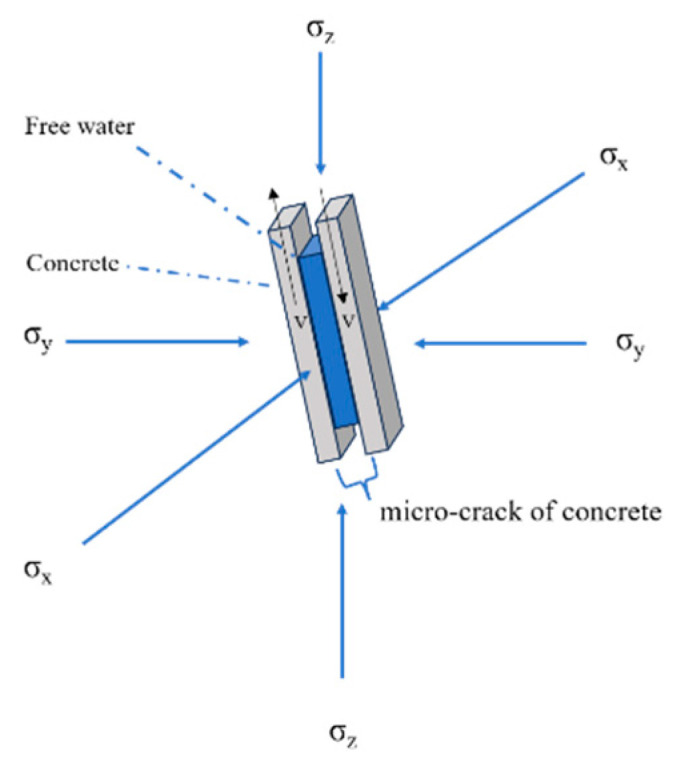
Weakening of the Stefan effect induced by confinement.

**Figure 11 materials-16-06591-f011:**
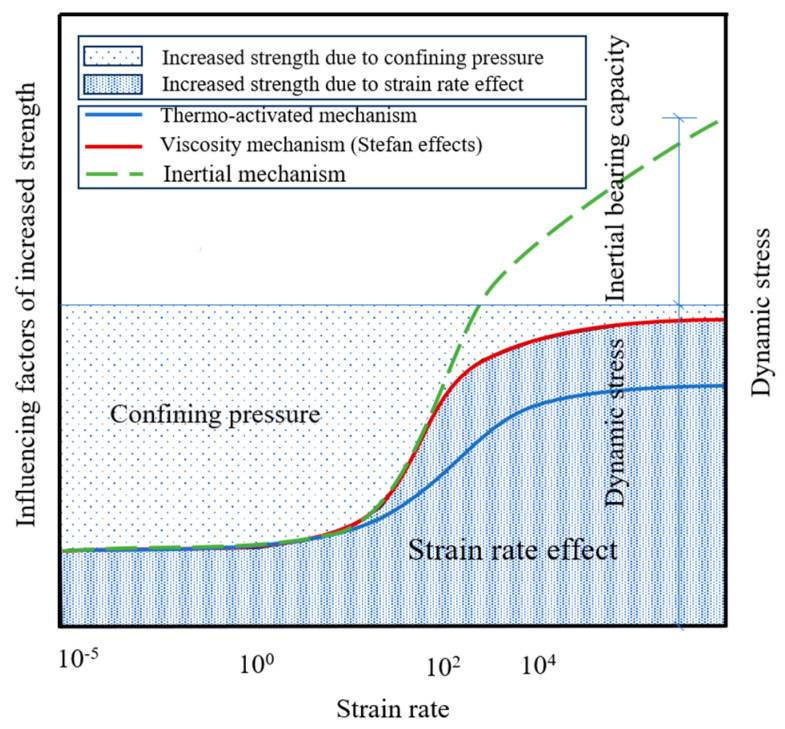
Mechanism of confining pressure and strain rate effects of concrete.

**Figure 12 materials-16-06591-f012:**
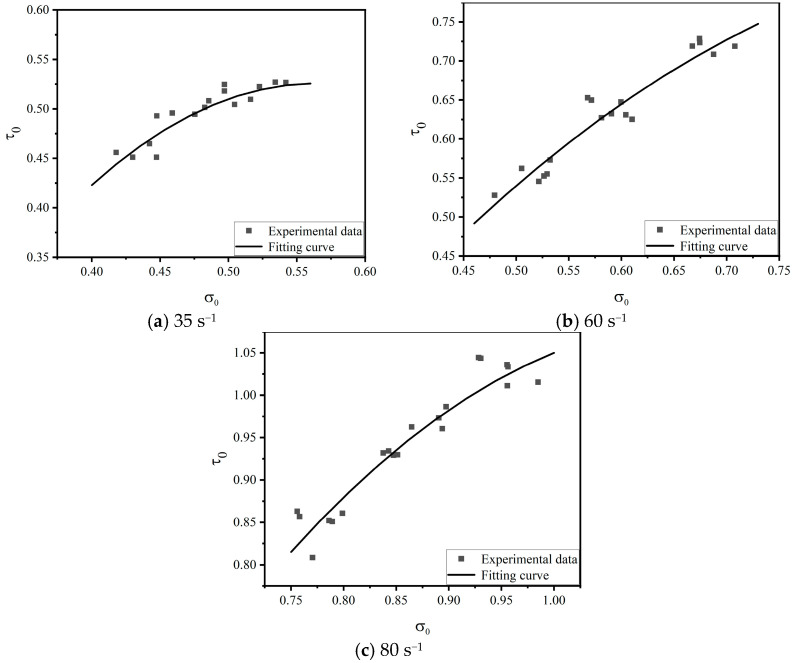
True triaxial dynamic failure criterion in octahedral stress space.

**Table 1 materials-16-06591-t001:** Mix proportions of concrete (unit: kg/m^3^).

	Cement	Silica Fume	Fine Aggregate	Superplasticizer	Water	Steel Fibers
U0.5	850	200	1000	40	150	39
U1	850	200	1000	40	150	79
U2	850	200	1000	40	150	158

**Table 2 materials-16-06591-t002:** UHPC’s 28 d compressive strength (unit: MPa).

	Measured Strength			Average Strength
U0.5	110.4	110.0	115.6	112.0
U1	126.4	130.5	131.0	129.3
U2	146.4	150.2	149.8	148.8

**Table 3 materials-16-06591-t003:** Mechanical properties of UHPC under true triaxial dynamic compression.

Sample No.	Strain Rate (s^−1^)	Confining Pressure (MPa)			Dynamic Pressure (MPa)		
		x	y	z	x	y	z
U0.5	~35	5.0	5.0	5.0	124.8	7.2	5.4
		7.5	7.5	7.5	118.8	9.3	10.0
		10.0	10.0	10.0	119.1	11.0	8.1
	~60	5.0	5.0	5.0	183.6	8.9	8.3
		7.5	7.5	7.5	181.7	11.2	9.4
		10.0	10.0	10.0	183.1	13.9	10.8
	~80	5.0	5.0	5.0	257.2	14.7	15.9
		7.5	7.5	7.5	263.6	17.4	18.1
		10.0	10.0	10.0	264.4	22.5	17.3
U1	~35	5.0	5.0	5.0	138.0	7.8	7.8
		7.5	7.5	7.5	143.2	7.8	9.86
		10.0	10.0	10.0	144.9	8.5	11.8
	~60	5.0	5.0	5.0	187.7	10.2	8.8
		7.5	7.5	7.5	188.4	10.9	10.8
		10.0	10.0	10.0	190.7	11.7	14.0
	~80	5.0	5.0	5.0	273.7	15.8	15.7
		7.5	7.5	7.5	272.3	14.9	17.2
		10.0	10.0	10.0	283.1	16.2	16.1
U2	~35	5.0	5.0	5.0	153.1	7.3	11.1
		7.5	7.5	7.5	151.4	9.8	8.2
		10.0	10.0	10.0	154.1	14.5	8.9
	~60	5.0	5.0	5.0	188.5	11.4	10.6
		7.5	7.5	7.5	192.3	11.8	11.0
		10.0	10.0	10.0	185.6	12.1	8.6
	~80	5.0	5.0	5.0	288.1	16.8	18.5
		7.5	7.5	7.5	288.8	19.4	20.2
		10.0	10.0	10.0	290.0	18.6	18.0

**Table 4 materials-16-06591-t004:** DIF(DIF_C_) empirical formula.

Refs	DIF(DIF_c_) Relations
CEB mode [34]	$DIF=\left\{\begin{array}{l} (\overset{•}{\varepsilon }/{\overset{•}{\varepsilon }}_{0}{)}^{0.014}\;\;\;\;\;\;\;for\;\overset{•}{\varepsilon }\leq 30\;s^{-1}\\ 0.012(\overset{•}{\varepsilon }/{\overset{•}{\varepsilon }}_{0}{)}^{1/3}\;\;\;\;\;for\;\overset{•}{\varepsilon }\leq 30\;s^{-1} \end{array}\right.$
Tedesco & Ross [35]	$DIF=\left\{\begin{array}{l} 0.00965\log\overset{•}{\varepsilon }+1.058\geq 1.0\;\;\;\;\;for\;\overset{•}{\varepsilon }\leq 63.1\;s^{-1}\\ 0.758\log\overset{•}{\varepsilon }-0.289\leq 2.5\;\;\;\;\;\;for\;\overset{•}{\varepsilon }>63.1\;s^{-1} \end{array}\right.$
Li & Meng [22]	$DIF=\left\{\begin{array}{l} 1+0.03438(\log\overset{•}{\varepsilon }+3)\;\;\;\;\;\;\;\;\;\;for\;\overset{•}{\varepsilon }\leq 10^{2}\;s^{-1}\\ 1.729{\left(\log\overset{•}{\varepsilon }\right)}^{2}-7.1372\log\overset{•}{\varepsilon }+8.5303\;\;\;\;\;for\;\overset{•}{\varepsilon }\geq 10^{2}\;s^{-1} \end{array}\right.$
Xu & Shan [26]	$DIF_{c}=\left\{\begin{array}{l} 1+0.02192(\log\overset{•}{\varepsilon }+3.771)\;\;\;\;\;\;\;\;for\;\overset{•}{\varepsilon }\leq 25.7\;s^{-1}\\ 2.147{\left(\log\overset{•}{\varepsilon }\right)}^{2}-5.408\log\overset{•}{\varepsilon }+4.466\;\;\;\;\;for\;\overset{•}{\varepsilon }>25.7\;s^{-1} \end{array}\right.$
Ren & Wu [33]	$DIF=\left\{\begin{array}{l} \;\;\;\;\;\;\;\;\;(\overset{•}{\varepsilon }/{\overset{•}{\varepsilon_{0})}}^{0.014}\;\;\;\;\;\;\;\;\;\;\;\;\;for\;\overset{•}{\varepsilon }\leq 30\;s^{-1}\\ 0.5835\log{\left(\overset{•}{\varepsilon }\right)}^{2}-1.5905\log\overset{•}{\varepsilon }+2.1988\;\;\;\;\;for\;\overset{•}{\varepsilon }>30\;s^{-1} \end{array}\right.$
Present study	$DIF_{c}=\left\{\begin{array}{l} \;\;\;\;\;\;\;\;\;(\overset{•}{\varepsilon }/{\overset{•}{\varepsilon_{0})}}^{0.014}\;\;\;\;\;\;\;\;\;\;\;\;\;for\;\overset{•}{\varepsilon }\leq 30\;s^{-1}\\ 1.7883\log{\left(\overset{•}{\varepsilon }\right)}^{2}-3.7413\log\overset{•}{\varepsilon }+2.5824\;\;\;\;\;for\;\overset{•}{\varepsilon }>30\;s^{-1} \end{array}\right.$

**Table 5 materials-16-06591-t005:** Values of fitting parameters under different strain rates.

Strain Rate/s^−1^	a	b	c	R^2^
35	−0.69	−4.31	−3.82	0.84
60	−0.32	−2.28	−1.12	0.91
80	−1.21	−4.02	−1.76	0.91

## Data Availability

Not applicable.

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
