# Peer review of "Study on the Dynamic Mechanical Properties of Ultrahigh-Performance Concrete under Triaxial Constraints"

_materials, 2023, doi:10.3390/ma16196591_

Round 1

Reviewer 1 Report

Review report: Study on the dynamic mechanical properties of UHPC under triaxial constraints. The detail comments are listed below:

1.       Abstract: Omit the unnecessary information and add some results at the end of the section.

2.       Introduction: Try to make a bridge between current and previous work. Refer to some recently published work. Also shorten the length of this section.

3.       Novelty and application: Add a separate section for novelty and application of work.

4.       Materials and methods: Section is presented well but need some corrections. Add a detail of experimental set up instead of a schematic image. Add the standard and image of specimen prepared for testing. How was the strength calculated?

5.       Results and Discussion: The major corrections are listed below:

a.        Add references for each equation.

b.       The role of strain rate should be discussed in detail.

c.        Strain Vs. time plot is not clear from discussion.

d.       Technical discussion need strengthening.

6.       Shorten the length of the conclusion section.

7.       References are ok. 

NA

Reviewer 2 Report

I congratulate the authors on an interesting scientific work that will certainly affect the development of the field of dynamic testing of the mechanical properties of UHPC for triaxial constraints.

The conclusion of the authors that dynamic axial stresses are characterized by high sensitivity to the strain rate should be emphasized.

To validate the effect of confinement pressure on the dynamic mechanical behavior of ultra-high-strength concrete (UHPC), this study used a true Hopkinson triaxial compression bar test system to perform UHPC dynamic compression tests under triaxial constraints. Triaxial compressive pressure was exerted on the sample by means of six steel rods controlled by three independent servo-hydraulic actuators. The test results show that the peak axial dynamic stress, peak lateral stress and peak axial strain of the UHPC are highly sensitive to strain rate under closed conditions.

Generally, I do not have substantive comments to the article and the research results obtained, but I am asking the authors for an answer whether, in their opinion, the study conducted on only 9 samples is not too little to reliably assess the research results?

Reviewer 3 Report

Authors are recommended to revise or explain the following aspects:

- the motivation from the introduction, lines 25 to 35, is partially true because the phenomenon of triaxial compression, researched in the work, is only theoretical, being impossible to be identified in reality. The fact that the action is static or dynamic is not relevant to the definition of the characteristic parameters regarding the behavior of the material, generally relevant is the moment of reaching yielding and exceeding the resistance of the material to a type of stress. When we refer to dynamic actions, we must take into account the specific residual deformations that overlap and accumulate with consequences in the occurrence of yielding at different action values than in the case of static application. The research carried out has the purpose of substantiating the values of some material parameters, such as Poisson's coefficients or identifying the behavior mechanism in the case of confinement, which involves preventing the development of specific linear deformations.

- in the introduction, the authors make confusion between the behavior of confined concrete to which confinement can be equated as an action of a pressure in the opposite direction that occurs due to the prevention of the development of specific linear deformations in the spatial state of tensions under the action of forces only on one or two directions. Poisson's coefficient states precisely this aspect: the ratio between the specific linear deformations that appear in two directions when the force acts in one direction. And preventing the occurrence of specific linear deformations after the second direction through confinement can be equated with a pressure of the opposite direction.

- the application of Hooke's law requires knowledge of the elasticity moduli, which are very important (for a homogeneous and isotropic material, the law is expressed: stress = strain x the elasticity modulus of the material in the direction of the load). In the paper, the authors do not refer to this characteristic (the modulus of elasticity) nor to how they characterized the researched material as: homogeneous or inhomogeneous, isotropic, orthotropic or anisotropic.

- as explained in the subsection. "2.2.2 Testing techniques" a dynamic shock-type action in the x direction is applied, but the dynamic action model is not detailed.

- the statements from lines 411-419 are confusing and have drafting errors.

- equations 14 and 15 as presented are confusing.

- the conclusions are confusing. For example. "UHPC is a strain rate-dependent material. As the strain rate increases, its peak stress, peak strain, equivalent peak stress, and equivalent peak strain all increase significantly; the confining pressure has little effect on the dynamic response in the x-axis direction but has an effect on the dynamic response in the x-axis direction. The influence of stress and strain along the y-axis and z-axis is greater."

Also:

- what is the motivation for using the word "true" in the expression "True triaxial test" or "true triaxial SHPB"?

- what means the ”true dynamic strength”?

- it is necessary to correct "s -1".

- figures 12 and 13 do not exist (lines 189 and 190).

- the style used in the expression in English is confused by repeating phrases, such as:

  "The passive confining pressure method is a passive confining pressure method implemented by adding a lateral confinement ring around the test piece in combination with the split Hopkinson bar device (SHPB), and it is an emerging technology that has been widely used in recent years.” (lines 102-105);

or

"Researchers generally believe that the inertia effect mainly controls the dynamic strength of concrete under a high strain rate. When concrete is in the high strain rate range, the macroscopic bearing capacity of concrete material increases with the increase in strain rate, while the true dynamic strength is at a very high strain rate. rate has a limit value. When the stress state is in the high strain rate range, the concrete can still bear the load, but the concrete already fails after the load is unloaded. Therefore, it was not considered within the scope of the present study." (lines 404-410).

There are many expressions of this type.

- in the figures there are explanations in Chinese.

Extensive editing of English language required (see the comments).

Reviewer 4 Report

The design of this research is on high level, detailed and well executed. The originality is not that high, but selection of samples and experimental techniques is still of very high quality. Unfortunately, in the current version, the manuscript is hard to read and follow.
As to the scientific content, I missed the analysis and discussion of changes strain curves in x, y and z directions. Also, the relation between dynamic equivalent stress and equivalent strian need more illustration. 

The quilty of the presentation needs to be enhanced espacially figs 5, 8 and 9 are hardly seen and follow.

The aauthors should add the uncertainties  for all data are presented in the current manucsript.
I suggest a major re-write with excluding all those technical mishaps (a native English writing colleague or agency might be useful) but also securing the "red line" of the story to become more clear and easier to follow. I also missed in conclusions remarks of the possible usefulness or applicability of such objects.

Moderate English correction is needed.

Round 2

Reviewer 3 Report

In the current version, the manuscript complies with the requirements. The authors responded to the comments submitted for the initial version.